# Effects of *Polygonatum sibiricum* on Physicochemical Properties, Biological Compounds, and Functionality of Fermented Soymilk

**DOI:** 10.3390/foods12142715

**Published:** 2023-07-15

**Authors:** Peng Wan, Han Liu, Yuanyuan Zhu, Haitao Xin, Yanli Ma, Zhizhou Chen

**Affiliations:** 1Zhang Zhongjing College of Chinese Medicine, Nanyang Institute of Technology, Nanyang 473000, China; wanpeng900522@163.com (P.W.); 15139008041@163.com (H.L.); 19836396659@163.com (Y.Z.); xht88400318@163.com (H.X.); 2College of Food Science and Technology, Hebei Agricultural University, Baoding 071000, China; chenzhizhou2003@126.com

**Keywords:** *P. sibiricum*, polysaccharides, fermented soymilk, physicochemical properties, isoflavones, antioxidant activity, organic acid

## Abstract

The purpose of this study was to investigate the effects of *Polygonatum sibiricum* (*P. sibiricum*) on microbial fermentation, physicochemical properties, and functional properties of fermented soymilk. Three types of fermented soymilk were prepared. The first type was fermented directly from regular soymilk (fermented soymilk, FSM), and the other two were fermented after adding *P. sibiricum* (*P. sibiricum* fermented soymilk, P-FSM) or *P. sibiricum* polysaccharides (*P. sibiricum* polysaccharides fermented soymilk, PP-FSM). The differences in physical and chemical indexes such as pH value, acidity, and water-holding capacity were mainly compared, and the differences in the contents of functional components such as total phenols, total flavonoids, soy isoflavones, γ-aminobutyric acid, and organic acids were compared. The functionalities of the three samples in terms of antioxidant activity were evaluated, and the relevance of each active substance was explored. Compared with the FSM group, the addition of *P. sibiricum* and *P. sibiricum* polysaccharides could not only significantly promote the fermentation of *Lactobacillus* but also significantly improve the stability of the finished products during storage and prolong the shelf life of the finished product. The conversion rates of glycoside soybean isoflavones in the PP-FSM and P-FSM groups were 73% and 69%, respectively, which were significantly higher than those in the FSM group (64%). At the end of fermentation, the γ-aminobutyric acid contents of the PP-FSM and P-FSM groups were 383.66 ± 1.41 mg/L and 386.27 ± 3.43 mg/L, respectively, while that of the FSM group was only 288.66 ± 3.94 mg/L. There were also great differences in the content and types of organic acids among the three samples, especially lactic acid and acetic acid. By comparing the antioxidant capacity of DPPH (1,1-Diphenyl-2-picrylhydrazyl free radical), AB-TS (2,2′-Azinobis-3-ethylbenzthiazoline-6-sulphonate), and iron chelation, it was found that both PP-FSM and P-FSM were superior to FSM, and the antioxidant capacity had a certain correlation with the contents of total phenols and total flavonoids.

## 1. Introduction

Soy is a high-quality vegetable protein free of lactose and cholesterol. It is rich in polyunsaturated fatty acids and isoflavones [1]. Fermented soymilk is a product made from soybean as the main raw material and fermented by *Lactobacillus*. Studies have reported that the fermentation process not only changes the physicochemical properties of soymilk but also enriches the sensory properties of soy milk [2]. For example, compared with unfermented soymilk, fermented soymilk contains more bioactive compounds, including γ-aminobutyric acid (GABA), organic acids (OA), and soybean isoflavones [3]. It has been demonstrated that fermented soymilk exhibits various bioactivities, such as antioxidant, anti-diabetic, anti-atherosclerotic, anticancer, anti-inflammatory, and immunomodulatory properties [4]. Numerous studies have proven that soy isoflavones have greater potential to regulate hormones, prevent cardiovascular disease, lower blood glucose and blood lipid, and increase antioxidant capacity [5,6]. It has been found that soymilk with GABA beverages could improve cell viability and significantly attenuate the release of lactate dehydrogenase, with potential neuroprotective activity [7]. At the same time, microorganisms can eliminate harmful substances and antioxidant factors in soymilk and also produce organic acids, peptides, alcohols, esters, and other aromatic substances to overcome the fishy taste of soybeans while giving the product an excellent fermented taste [8].

Chinese traditional medicine has a large number of botanicals of the same origin as food and medicine. As a traditional Chinese medicine, *P. sibiricum* has been used for 2000 years [9]. It contains a variety of functional ingredients, such as polysaccharides, flavonoids, saponins, alkaloids, amino acids, anthraquinones, and microelements [10]. The polysaccharide is the highest component among them, and it has high biological activity [11]. *P. sibiricum* has been demonstrated to exhibit diverse biological activities, including hypoglycemic and lipid-lowering, antioxidant [12], anti-inflammatory, anticancer [13], regulation of gut microbiota [14], anti-osteoporosis [15], and neuroprotection [16]. At present, *P. sibiricum* can be used for diabetes, coronary heart disease, hyperlipidemia, and other clinical treatment [17]. *P. sibiricum* has high economic value because of its medicine and food homology, and a series of beverages, wines, and yogurt products developed with it as raw materials have gradually entered the market [18]. The researchers found that natural medicinal and edible plants can increase the extraction rate of effective components after probiotic fermentation and produce new components that are more easily absorbed, more active, and with fewer side effects [19]. For example, the yield of polysaccharides of *P. sibiricum* was increased by 10% after microbial fermentation. At the same time, the raw product stimulation and numb-tongue sensation were removed, and the antioxidant and anti-fatigue effects were improved [20].

However, there is little information on the use of *P. sibiricum* or *P. sibiricum* polysaccharides as functional ingredients for fermented soymilk manufacturing. Three kinds of functional soybean milk solid beverage with combined fermentation were prepared in this study, respectively: *P. sibiricum* polysaccharide fermented soymilk of (PP-FSM), *P. sibiricum* fermented soymilk (P-FSM), and regular fermented soymilk (FSM). By comparison, the effects of adding *P. sibiricum* or *P. sibiricum* polysaccharide on microbial fermentation, physicochemical properties, and functional properties of fermented soybean milk were studied. At first, the fermentation features, such as the viable cell count, pH, titratable acidity, water-holding capacity, and viscosity of soymilk during fermentation and storage, were investigated. Then, variations of functional compounds, including total phenols, total flavonoids, soy isoflavones, GABA, OA, and other active components of three kinds of fermented soymilk during fermentation and storage, were also detected. Finally, the antioxidant activities of three fermented soymilks were evaluated by measuring the free-radical-scavenging activity of the ABTS, DPPH, FRAP, and total antioxidant activity.

## 2. Materials and Methods

### 2.1. Materials

The P. sibiricum and soybeans were obtained from Lianyuan Biotechnology Co., LTD (Nanyang, Henan, China). All chemicals used were analytical grade, and purchased from Shanghai Maclin Biochemical Technology Co., LTD (Shanghai, China). Methanol and acetonitrile are chromatographically pure, purchased from Shanghai Aladdin Biochemical Technology Co., LTD (Shanghai, China). All standard products were purchased from Shanghai Yuanye Biotechnology Co., LTD (Shanghai, China).

### 2.2. Preparation of P. sibiricum Extract and P. sibiricum Polysaccharide

The extraction method was based on previous studies and modified slightly [21]. The rhizome of *P. sibiricum* was dried at 40 °C for 48 h, crushed by a micro pulverizer, and sieved through a 60-mesh sieve to obtain *P. sibiricum* powder. The *P. sibiricum* powder was soaked in distilled water (1:25, wt/vol) for 30 min, and 3.48% cellulase and papain (the ratio was 20:9, wt/wt) were added. Ultrasonic-assisted enzymatic digestion extraction was performed at 50 °C for 40 min and then immediately put in a boiling water bath for 10 min to inactivate the enzyme. Then, the mixture was extracted in hot water twice at 90 °C for 4 h, and the supernatants were combined after centrifugation. The supernatants were concentrated by rotary evaporator and then lyophilized by vacuum freeze-dryer to afford the crude extract of *P. sibiricum* (consisted of 28.22 ± 0.325% polysaccharide, 17.48 ± 0.524% water-soluble protein, 6.02 ± 0.089% total saponins, 3.25 ± 0.12% total flavonoids, and 5.31 ± 0.693% total phenols). The supernatants were concentrated and precipitated with four times volume of anhydrous ethanol at 4 °C overnight. The precipitates were collected by centrifugation at 10,000 rpm for 15 min and washed with anhydrous ethanol. Then, the precipitates were re-dissolved in distilled water and lyophilized to yield the crude polysaccharides of *P. sibiricum*.

### 2.3. Fermentation of Soymilk

The cleaned soybeans were blanched at 90 °C for 5 min and soaked in 0.25% NaHCO_3_ solution for 10–12 h. The soybeans were rinsed to neutral with water and then were mixed with boiling water in a ratio of 1:6 (wt/vol). Reconstituted milk and soy milk were mixed at a ratio of 4:6, adding sugar (6%) and stabilizers (0.1% carrageenan and 0.1% pectin) to obtain soymilk mixture, adding 1% fermented strain (*Streptococcus* thermophilus: *Lactobacillus* bulgaricus = 1:1), then incubating at 42 °C for 6.5 h to create regular fermented soymilk (FSM). Soymilk mixture was inoculated by adding 2% extract powder of *P. sibiricum,* namely P-FSM, or polysaccharides of *P. sibiricum* (PP-FSM). All samples were prepared in triplicate.

In order to dynamically detect the physicochemical properties, microbiological determination, and functional activity of fermented soymilk, we chose to collect samples every 1 h during the fermentation process. In the storage period, samples were collected on the 1st day, 4th day, 7th day, 15th day, and 20th day.

### 2.4. Determination of Physicochemical Properties

The pH value was measured by a pH meter, and the titratable acidity (TA) was analyzed using a titration method with 0.10 M sodium hydroxide [22].

The water-holding capacity (WHC) was based on previous studies and modified slightly [23]. The mass of the empty centrifuge tube was recorded as *Z*. After adding the sample, the total mass of the empty tube and the sample was recorded as *Z*1. Then, the tube containing the sample was centrifuged at 7500 rpm/min for 5 min. After discarding the supernatant, the total mass of the precipitate and the empty tube was recorded as *Z*2. WHC was calculated according to the following formula:$$WHC\left(\%\right)=\frac{\left(Z1-Z\right)}{\left(Z2-Z\right)}\times 100.$$

The sample of viscosity was measured using a Brookfield RV-D-II + pro viscometer (Brookfield Engineering Laboratories INC., Middleboro, MA, USA) [24].

### 2.5. Determination of Microbial Viability

The determination of microbial viability was performed according to the Chinese national standard (General Administration of Quality Supervision, Inspection and Quarantine of China & Standardization Administration of China, 2011). The number of viable bacteria was determined by plate-pouring method, and the colonies were counted after 48 h incubation at 36 °C.

### 2.6. Active Substances

#### 2.6.1. Determination of Total Phenolic and Total Flavonoid

The total phenolic content (TPC) was determined with Folin–Ciocalteu phenolic study according to a method [25] with minor modifications. The aluminum chloride (AlCl_3_) colorimetric assay method was used to determine the total flavonoid content (TPF) [26].

#### 2.6.2. Determination of Isoflavones

Isoflavones were determined using high-performance liquid chromatography (HPLC) according to the cited method with slight modifications [27]. The samples (7.5 g) were dissolved in 25 mL of 90% methanol and incubated in an ultrasonic cleaner at 60 °C for 1 h. Then, the mixture was centrifuged at 4000 rpm for 30 min. The supernatant was filtered through a 0.22 μm filter prior to the HPLC assay. An Agilent ZORBAX Eclipse Plus C18 column (4.6 mm × 250 mm, 5 μm) was employed for chromatographic separation at 40 °C. Elution solution consisted of solvent A (0.1% acetic acid solution) and solvent B (0.1% acetonitrile acetate solution). The gradient elution program was set with a flow rate of 1 mL/min, and UV detection (1290 Infinity II diode array detector) was performed at 260 nm.

#### 2.6.3. Determination of GABA

The content of GABA was determined by HPLC with reference to previous methods and modifications [28]. The GABA extracts were prepared with 5 g of samples from each species and homogenized with 10 mL of 7% TCA solution for 2 h in a water bath at 40 °C. Then, the mixture was centrifuged at 10,000 rpm for 10 min. A total of 2 mL of supernatant was mixed with 2 mL ethanol left at 4 °C for 1 h to remove the protein and polysaccharides. The mixture was centrifuged at 10,000 rpm for 10 min. The supernatant was derivatized with phenyl isothiocyanate reagent and filtered through a 0.22 μm filter paper prior to injection. An Agilent ZORBAX Eclipse Plus C18 column (4.6 mm × 250 mm, 5 μm) was employed for chromatographic separation at 40 °C. Elution solution consisted of solvent A (sodium acetateanhydrous: triethylamine: water = 1.64:0.5:997.86) and solvent B (acetonitrile: water = 8:2). The gradient elution program was set with a flow rate of 1 mL/min, and UV detection (1290 Infinity II diode array detector) was performed at 254 nm.

#### 2.6.4. Determination of OA

The content of OA was measured by HPLC according to a previous method with some modifications [29]. The samples (1.2 g) were added with 8 mL 0.01 mol/L sulfuric acid solution and shaken for 1 min. The supernatant was filtered through 0.22 μm membrane after centrifugation at 10,000 rpm for 10 min. The chromatographic separation was performed on an Agilent ZORBAX Eclipse Plus C18 column (4.6 mm × 250 mm, 5 μm). Elution solution was 0.02 mol/L dibasic sodium phosphate solution (phosphoric acid regulates pH to 2.1). The gradient elution program was set as follows: the flow rate was 0.9 mL/min, the detection temperature was 35 °C, and the UV detection (1290 Infinity II diode array detector) was performed at 210 nm.

### 2.7. Antioxidant Activity

The antioxidant activity of the samples was assessed by measuring the free-radical-scavenging activity of the ABTS+, DPPH, FRAP, and total antioxidant activity.

Total antioxidant activity: the sample was centrifuged at 10,000 r/min for 10 min. The supernatant (0.3 g) was diluted in a 10 mL stoppered tube by adding absolute methanol (0.7 mL). Then, 1 mol/L HCl solution (0.2 mL), 1% K_3_[Fe (CN)_6_] solution (1.5 mL), 1% SDS solution (0.5 mL), and 0.2% FeCl_3_ solution (0.5 mL) were added successively. The absorbance value was measured at 750 nm after standing in distilled water for 30 min at room temperature. The VC was used as a positive control.

ABTS+ free-radical-scavenging activity [30]: equal volumes of 7 mmol/L ABTS+ solution and 2.5 mmol/L potassium persulfate solution were mixed, and the reaction was kept away from light for 15 h. The absorbance was diluted with PBS solution pH 7.4 at 734 nm to 0.700 ± 0.005. The sample (0.2 mL) supernatant was mixed with ABTS+ solution (7.8 mL), and the reaction was kept away from light for 10 min. The absorbance was measured at 734 nm, and the experiment was repeated three times. The results of the antioxidant activity of samples are expressed as a percentage of inhibition (%).

DPPH free-radical-scavenging activity [31]: the supernatant (0.2 mL) was added to an equal volume of 0.2 mmol/L DPPH radical ethanol solution (stored away from light). The mixture was reacted for 30 min at room temperature in a dark environment. The absorbance was measured at 517 nm, and the experiment was repeated three times. The results of the antioxidant activity of samples are expressed as a percentage of inhibition (%).

FRAP free-radical-scavenging activity: the determination method of Fe^2+^ chelating ability was derived according to previous research [32].

SPSS 22.0 software was used to analyze the correlation between various active substances and antioxidant capacity.

### 2.8. Statistical Analysis

The statistical analysis was performed using Origin 2021 software (Origin Lab Co., Ltd.; Northampton, MA, USA). Duncan’s multiple range test and one-way ANOVA variance analysis were performed to compare significant differences (*p* < 0.05) using SPSS 22.0 for Windows (IBM). All experiments were performed in triplicates, and the data were expressed as mean ± standard deviation.

## 3. Results and Discussion

### 3.1. Physicochemical Properties

The TA and pH are important indicators used to measure the quality of soymilk during fermentation and storage. The value of TA and pH can reflect the growth status and activity of *Lactobacillus* in soymilk, which are important indexes to determine whether soymilk reaches the end of fermentation. Generally, the pH value of fermented dairy products should be kept above 4.00 during the shelf life [33]. The physicochemical characteristics of three types of fermented soymilk are summarized in Figure 1A. The TA, pH, and WHC values of three kinds of fermented soymilk showed a trend of significant increase followed by a slow increase under the constant temperature fermentation environment of 42 °C. The pH values and TA values of PP-FSM samples decreased significantly from 1 h to 4 h, while the pH values and TA values of P-FSM and FSM samples varied significantly from 2 h to 4 h. The fermentation time of the three samples was about 6.5 h. At the end of fermentation, the pH values and TA values of PP-FSM, P-FSM, and FSM samples were 4.20 ± 0.06, 4.32 ± 0.15, and 4.38 ± 0.12 and 95.00 ± 2.64 °T, 86.62 ± 1.36 °T, and 75.56 ± 0.60 °T, respectively. In the comparative study, it was found that the TA values of PP-FSM and P-FSM samples were significantly higher than those of FSM during fermentation (*p* < 0.05). The highest increasing trend was observed in the samples with the addition of the *P. sibiricum* polysaccharide. It may be that the *P. sibiricum* polysaccharide has a beneficial biogenic effect on the growth of microorganisms. It promotes the fermentation of lactic acid bacteria and accelerates the accumulation of organic acids, thus enhancing the acidity of soymilk [34].

Excessive acidification is an undesirable characteristic of fermented dairy products during the storage period [35]. During the storage times, the TA values of PP-FSM and P-FSM were stable, and the degree of acidification was not severe. On the contrary, the TA values and pH values of FSM varied significantly, and the quality of FSM continued to decrease. In the process of sample storage, *P. sibiricum* can reduce the degree of post-acidification and prolong the shelf life of products.

The WHC and viscosity are important indexes used to evaluate the tissue status of fermented soymilk. Low WHC can cause whey precipitation during the storage period and seriously affect the texture or flavor of finished products. The WHC of the three samples changed significantly during the fermentation (Figure 1B). The WHC of PP-FSM samples increased from 17.09 ± 1.30% to 78.28 ± 1.35%, the P-FSM samples improved from 18.90 ± 1.29% to 66.34 ± 1.36%, and the FSM samples enhanced from 15.06 ± 1.42% to 64.85 ± 0.89%. It was speculated that the significant variation in PP-FSM was related to the adsorption of polysaccharides. In the process of fermentation, the polysaccharide interacted with protein molecules in the form of chemical bonds in the gel structure of soymilk, which enhances the gel structure of the system [36,37], effectively retains water, and prevents whey precipitation. Compared with the end of fermentation, the WHC of the three samples decreased by 2.00%, 6.61%, and 13.36% during the storage times, respectively. It can be seen that the WHC of PP-FSM was significantly more stable than that of the other two samples. The results showed that the addition of extract and polysaccharide of *P. sibiricum* was beneficial to the texture stability and flavor retention of the fermented soymilk during shelf life. On the one hand, the apparent viscosity reflects the degree of *Lactobacillus* fermentation; on the other hand, it has a great influence on the microstructure and taste of the fermented soymilk. As the WHC of the samples increased, the viscosity also gradually increased and stabilized (Figure 1C). Significant differences were found between the three samples at the end of fermentation. After 15 days of storage, the viscosity of the soymilk decreased. This may be due to the destruction of the network structure formed inside, which caused the whey to precipitate out and the viscosity to decrease [24]. Many esters among them have different saturation in water under different conditions, so partial emulsion separation will occur. These reasons can lead to a certain degree of viscosity reduction.

### 3.2. Microbial Viability

Figure 2 shows the viable bacteria counts of PP-FSM, P-FSM, and FSM during fermentation and preservation. According to the Chinese standard, the number of *Lactobacillus* viable at the completion of soymilk fermentation is ≥1 × 10^6^ CFU/mL. When fermentation was completed, the viable bacteria counts of the three groups were 8.74 × 10 CFU/mL, 8.57 × 10^8^ CFU/mL, and 8.33 × 10^8^ CFU/mL, respectively. The results showed that all samples satisfied the above condition. The whole fermentation process was divided into three stages: per-fermentation, mid-fermentation, and post-fermentation. In the early stage of fermentation, *Streptococcus thermophilus* mainly produced acid. With the decrease in pH value, *Lactobacillus bulgaricus* started to grow logarithmically, which was the middle stage of fermentation. In the later stage of fermentation, both of them entered the stable stage, and the rate of acid production gradually slowed [38]. The number of *Lactobacillus* increased significantly during the fermentation of three kinds of fermented soymilk. The growth rate was the highest at 0–4 h, which is in the logarithmic growth period. The viable bacteria counts of PP-FSM and P-FSM were slightly higher than those of FSM, which might be attributed to the *P. sibiricum* polysaccharide that could promote *Lactobacillus* to better utilize nutrients in soymilk. However, it was also indicated that too much polysaccharide would form a hinge structure and reduce the water activity of the system [39], which was not conducive to the growth of *Lactobacillus*.

The mortality rate of FSM was the highest during the storage period. It has been reported that the polysaccharide of *Polygonatum* can inhibit the growth of harmful bacteria such as *Escherichia coli* and *Staphylococcus aureus* [40]. We speculated that the PP-FSM and FSM had fewer competitive harmful bacteria, so the viable bacteria count of two kinds of fermented soymilk was always higher than that of FSM during storage.

### 3.3. Isoflavones

Isoflavones are a secondary metabolite in soybean growth, which have beneficial effects such as regulating estrogen levels, reducing blood glucose and lipids, preventing cardiovascular and cerebrovascular diseases, and antioxidation. Previous research has also mentioned that six kinds of isoflavones are exhibited in soybeans, which mainly exist in the form of glycosidic isoflavones (daidzin, genistin, glycitin) and account for 97–98% [41]. It also includes aglycone types such as daidzein, glycitein, and genistein. The results showed that the glycosidic isoflavones cannot easily be absorbed due to their being highly binding to the sugar moiety [42]. During the fermentation process, the compositions and contents of isoflavones change under the action of β-glucosidase produced by microorganisms, which converts the glycosidic isoflavones into aglycone isoflavones that are more easily absorbed by the body [43].

The contents of isoflavones in different groups during fermentation and storage are shown in Figure 3. The results indicated that fermentation could promote the conversion of isoflavones glycosides to aglycones. As shown in Figure 3A–C, both daidzin and genistin decreased significantly in three samples, which confirmed they were transformed into daidzein and genistein. The content of daidzin in PP-FSM, P-FSM, and FSM groups decreased from 70.35 ± 0.69 mg/kg, 69.09 ± 1.34 mg/kg, and 68.68 ± 1.67 mg/kg to 12.84 ± 1.03 mg/kg, 15.89 ± 1.04 mg/kg, and 20.84 ± 1.01 mg/kg, respectively. The conversion of PP-FSM and P-FSM was higher than that of FSM by 12.5% and 7.75%, respectively. The conversion rates of genistin were 80.31%, 74.15%, and 69.84%, respectively. Obviously, the sample with *P. sibiricum* was higher than that of ordinary fermented soymilk. The content of glycitin is depicted in Figure 4B. The contents of fermented soymilk did not change significantly.

The content changes of aglycone isoflavones are shown in Figure 3D–F. The content of aglycone isoflavones increased after fermentation. The content of daidzein in PP-FSM, P-FSM, and FSM groups improved from 28.88 ± 0.96 mg/kg, 28.11 ± 1.80 mg/kg, and 28.96 ± 0.93 mg/kg to 72.80 ± 0.20 mg/kg, 68.30 ± 0.96 mg/kg, and 62.88 ± 0.96 mg/kg. After fermentation, the content of glycitein of FSM increased by about 2 times, while the sample of PP-FSM and P-FSM increased by about 2.5 times. At the end of fermentation, the genistein content in PP-FSM and P-FSM was 30.27 ± 0.89 mg/kg, 33.97 ± 0.51 mg/kg, while that in FSM was 25.33 ± 0.83 mg/kg.

In general, fermented soymilk supplemented with extract or polysaccharide of *P. sibiricum* had a higher conversion capacity of glycosidic isoflavones than regular fermented soymilk. Studies found that the addition of lactose can promote the growth of the strain, enhance the activity of β-glucosidase, and significantly improve the conversion rate of glycosidic isoflavones [44]. Therefore, it may be that the addition of *P. sibiricum* changes the carbon source composition in the fermentation process of *Lactobacillus*. Accordingly, the fermentation of Lactobacillus was more complete, and the conversion rate of glycosidic isoflavones was higher.

The contents of six isoflavones were detected during the storage period, and it was found that the three kinds of fermented soymilk had little change trend. This might be due to the low metabolic ability of *Lactobacillus* in the low-temperature environment, which inhibited the activity of β-glucosidase and caused no significant change in the isoflavone content.

### 3.4. GABA

GABA is the end product of glutamic acid decarboxylation (GAD) in *Lactobacillus*, which is synthesized from pyridoxal phosphate-dependent GAD through the irreversible α-decarboxylation of L-glutamic acid. It is mainly produced by the TCA cycle and glycolysis pathway [45]. It has been shown that GABA is the main inhibitory neurotransmitter in the vertebrate central nervous system and also in the brain, which has the effects of lowering blood pressure, regulating hormone secretion, treating epilepsy, and anti-arrhythmia [46].

Figure 4 presents the content of GABA in all groups. The content of GABA in the three groups showed a significant difference. The contents of GABA in the three samples were 75.93 ± 2.28 mg/L, 76.27 ± 2.96 mg/L, and 77.71 ± 3.18 mg/L at 0 h, respectively. It could be found that GABA content did not change significantly in the early stage of fermentation. The contents of GABA in PP-FSM and P-FSM samples at 3–6 h ranged significantly from 155.97 ± 1.25 mg/L and 154.27 ± 2.16 mg/L to 305.51 ± 1.52 mg/L and 304.24 ± 1.42 mg/L, respectively. However, the content of GABA in FSM samples was lower than that in PP-FSM and P-FSM samples. It was reported that GABA showed the highest activity in the pH range of 4.5–5.5 [47]. The pH of PP-FSM and P-FSM samples decreased below 5.5 firstly due to the prebiotic effect of *P. sibiricum* on *Lactobacillus* to promote their growth. At the end of fermentation, the GABA content of the three samples was 383.66 ± 1.41 mg/L,386.27 ± 3.43 mg/L, and 288.66 ± 3.94 mg/L, respectively. The GABA content of PP-FSM and P-FSM samples was significantly higher than that of FSM samples. As shown in Figure 4, the overall trend of GABA content of the three samples had no obvious change during storage. Only the FSM sample showed slight fluctuation, which may be caused by sample detection error. Studies have shown that the presence of glucose, lactose, and sucrose can produce large amounts of lactic acid to activate the synthesis of GABA and increase its production [7]. In this study, the addition of extract and polysaccharide of *P. sibiricum* changed the carbon source components in the fermentation system, which may be conducive to the synthesis of GABA, resulting in a change in GABA content. Therefore, it is necessary to further study how to control the addition amount and fermentation conditions of *P. sibiricum* to promote the production of GABA.

### 3.5. Organic Acid

*Lactobacillus* is a kind of chemotrophic heterotrophic microorganism whose metabolic growth needs carbohydrates to provide a carbon source. After lactose and citric acid conversion, the nutritional composition of soymilk changed. This process produces flavor compounds such as some carbonyl compounds and volatile and non-volatile acids. Among them, organic acids are the critical or precursor substances of sour milk flavor substances. The seven organic acids were detected in the samples—in this study, namely oxalic acid, malic acid, pyruvic acid, lactic acid, acetic acid, citric acid, and succinic acid. The main organic acid content changes in the three samples during fermentation and storage are shown in Figure 5.

The accumulation of lactic acid and acetic acid and the consumption of citric acid and malic acid during fermentation may be mainly related to the metabolism of acid-producing microorganisms. The content of lactic acid produced by *Lactobacillus* is different according to its fermentation type. It can be divided into two fermentation types: homolactic fermentation and heterotypic lactic fermentation. Homolactic fermentation can convert 80% of glucose to lactic acid through glycolysis. *Streptococcus* thermophilus and *Lactobacillus* bulgaricus belong to the former. The changes in lactic acid content in fermented soymilk samples during fermentation and storage are shown in Figure 5A. The content of lactic acid was significantly increased after fermentation. The samples in the PP-FSM group and P-FSM group were significantly higher than those in the FSM group at the end of the fermentation and storage period, reaching 9.55 ± 0.26 mg/g and 8.24 ± 0.27 mg/g, respectively. The citric acid, succinic acid, malic acid, and pyruvic acid content varied, as shown in Figure 5B–E. These four acids were the main organic acids involved in the TCA cycle. The citric acid content decreased significantly after fermentation, and the malic acid, pyruvic acid, and succinic acid contents increased. Citric acid, as the initial acid of the TCA cycle, was converted to organic acids such as malic acid, pyruvic acid, and succinic acid by a series of catalytic reactions under the action of three key enzymes of citrate synthase, isocitrate dehydrogenase, and α-ketoglutarate decarboxylase family [48]. As shown in Figure 5F,G, both oxalic acid and acetic acid content showed an increasing trend. Overall, it was observed that the samples of PP-FSM and P-FSM groups produced more organic acids. Therefore, the flavor, stability, and texture of these two samples were better than those of the FSM group.

### 3.6. TPC, TFC, and Antioxidant Activity

The TPC, TFC, and antioxidant activity of different samples at the end of fermentation were determined, as shown in Table 1. The TPC of the three groups was 2.59 ± 0.028 mg/g, 2.60 ± 0.057 mg/g, and 2.04 ± 0.023 mg/g, respectively. The TFC was 2.41 ± 0.28 mg/g, 2.10 ± 0.034 mg/g, and 1.86 ± 0.074 mg/g successively. The results showed that the TPC and TFC of the PP-FSM and P-FSM groups were significantly higher than those of FSM. When VC was used as a contrast, the scavenging ability of PP-FSM, P-FSM, and FSM on DPPH free radicals was 57.16 ± 0.33%, 57.63 ± 0.31%, and 52.67 ± 0.13%, respectively. The free-radical-scavenging activity of ABTS was increased by 10.75% and 3.31% with the addition of extract and polysaccharide of *P. sibiricum*, respectively. Compared with FSM, the chelating ability of iron ions improved to 40.89 ± 0.78% and 38.82 ± 0.28%. The TAP of the three groups were 257.93 ± 1.25 mg/g, 263.4 ± 1.67 mg/g, and 215.17 ± 0.98 mg/g, respectively. Both PP-FSM and P-FSM showed excellent antioxidant capacity. This was combined with the fact that the TPC and TFC in FSM were also lower than the other two groups, presumably related to the TPC and TFC in fermented soymilk. It may also be caused by the differences in the types and structures of phenols and flavonoids in different systems [49], which need to be further studied and discussed. Meanwhile, the active substances, such as polysaccharides and proteins, were increased after the addition of *P. sibiricum*, which further improved the antioxidant capacity of the samples [50].

The correlation analysis between the contents of total phenol, total flavonoid, organic acids, GABA, and isoflavones with various antioxidant indexes was discussed. The results are shown in Figure 6. In this study, the TFC and TPC showed an extremely significant positive correlation with the oxidation resistance index (*p* < 0.05). The results indicated that the TPC and TPC were closely related to the antioxidant activity of *P. sibiricum* fermented soymilk and were the main component of the antioxidant activity of the system. The phenol and flavonoid are the main material basis of antioxidant activity in plants, in which the neighboring phenolic hydroxyl groups in the phenolic hydroxyl structure can be easily oxidized to quinone structures. They have a strong ability to capture free radicals such as DPPH. The results of this study were similar to those of Hussein’s [51]. Among them, oxalic acid, pyruvic acid, and lactic acid showed a highly significant positive correlation with antioxidant activity (*p* < 0.05), while malic acid, acetic acid, citric acid, and succinic acid were not correlated with the antioxidant capacity. It was found that the scavenging effect of organic acids on free radicals depends on properties such as the molecular mass, number of aromatic rings, and hydroxyl substituents [52]. The content of GABA was highly significantly and positively correlated with antioxidant capacity (*p* < 0.05). The correlation between the six isoflavones and each antioxidant index was investigated. Daidzein, glycitein, and genistein were found to be significantly and positively correlated with the antioxidant capacity (*p* < 0.05). Daidzin and glycitin were highly significantly and negatively correlated with antioxidant properties (*p* < 0.05). However, there was no correlation between genistin and each antioxidant index.

## 4. Conclusions

This study showed that the physicochemical parameters, such as TA and WHC of PP-FSM and P-FSM, were better than those of FSM and relatively stable during storage. Compared with FSM, both have higher contents of active substances such as isoflavones, GABA, and organic acids, and also their antioxidant properties are superior. On the one hand, the addition of extract and polysaccharides of *P. sibiricum* can significantly improve the texture and flavor of the products and extend their shelf life through their antimicrobial effect. On the other hand, it can promote the production and transformation of some active substances, thus increasing the functionality of the products. At the same time, the combination of medical-food homologous materials with soymilk improved the functionality of fermented soymilk. The results of this study provide the practical basis for the development of deep-processed products and new functional foods of *P. sibiricum* and provide the theoretical basis for consumers to choose high-quality and nutrient-rich fermented soymilk.

## Figures and Tables

**Figure 1 foods-12-02715-f001:**
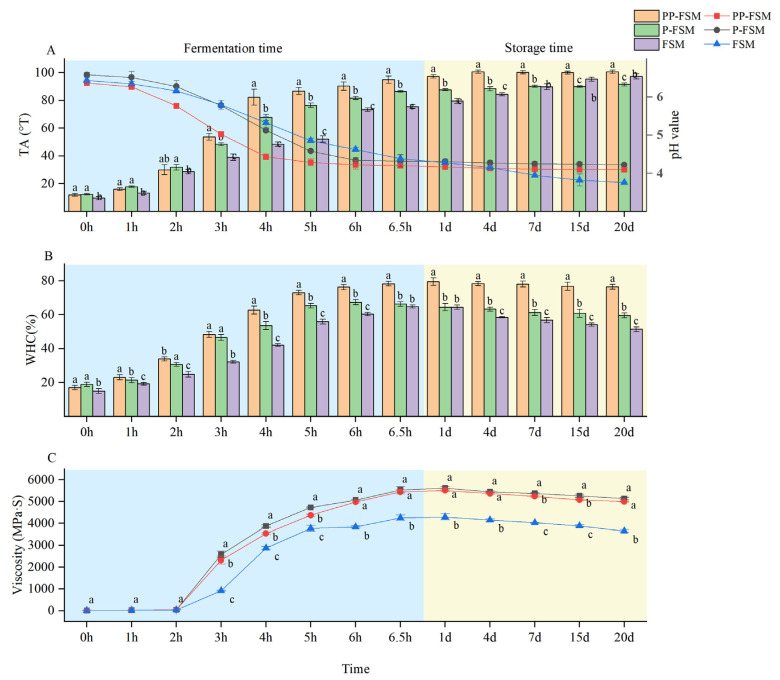
Comparison of physicochemical properties of fermented soymilk in different groups. (**A**) Changes in pH value and TA value of three samples. (**B**) Changes in WHC of three samples. (**C**) Changes in viscosity of three samples. a–c: Means within three samples with different superscripts were significantly different (*p* < 0.05).

**Figure 2 foods-12-02715-f002:**
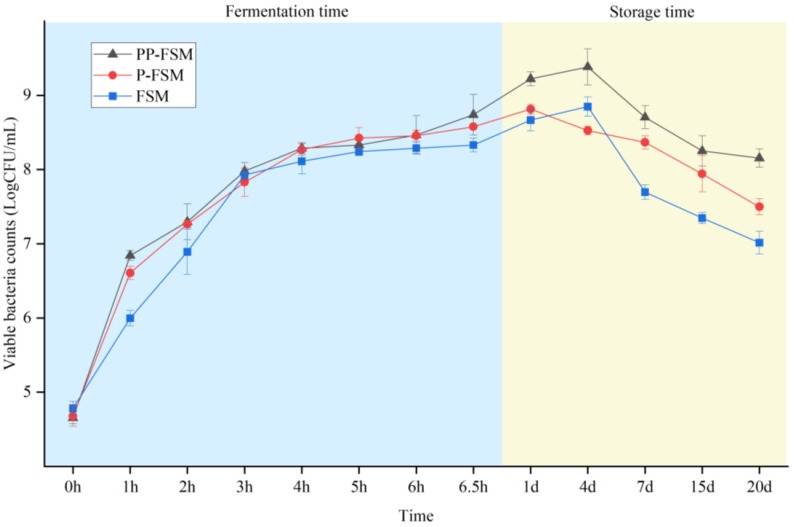
Viable bacteria counts of fermented soymilk in different groups.

**Figure 3 foods-12-02715-f003:**
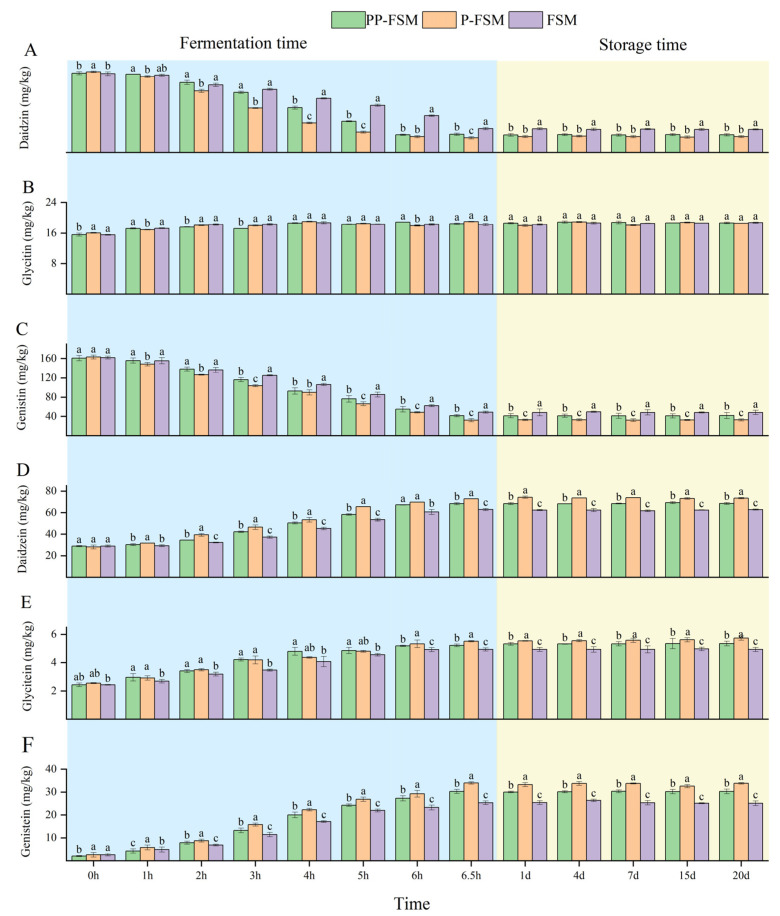
Isoflavone content distribution diagram in different fermented soymilk. (**A**) Daidzin. (**B**) Genistin. (**C**) Glycitin. (**D**) Daidzein. (**E**) Glycitein. (**F**) Genistein. a–c: Means within three samples with different superscripts were significantly different (*p* < 0.05).

**Figure 4 foods-12-02715-f004:**
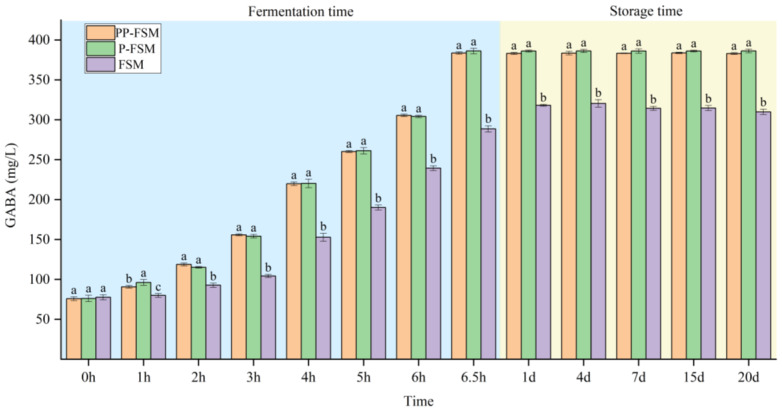
GABA contents in different fermented soymilk. a–c: Means within three samples with different superscripts were significantly different (*p* < 0.05).

**Figure 5 foods-12-02715-f005:**
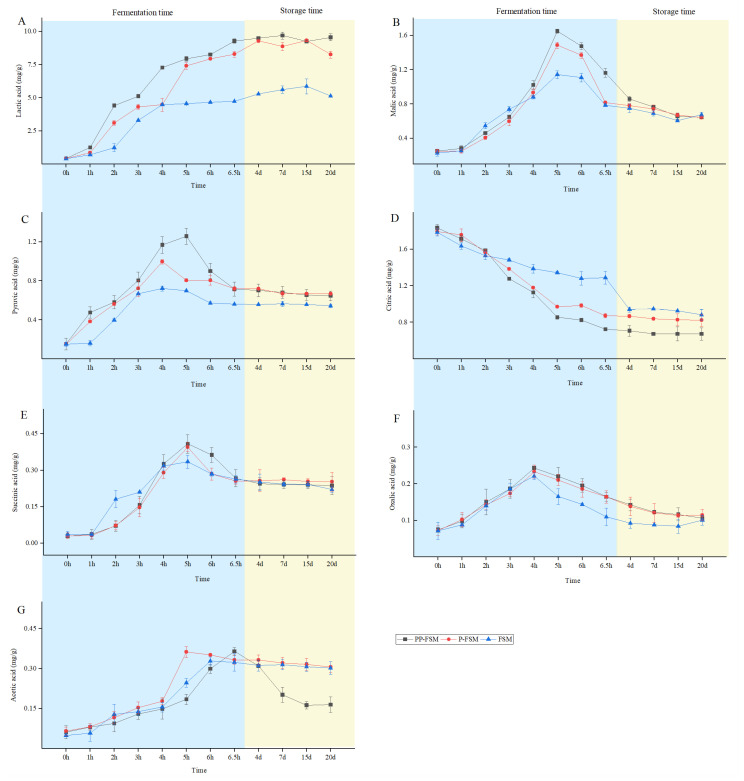
Organic acid contents in different fermented soymilk. (**A**) Lactic acid. (**B**) Malic acid. (**C**) Pyruvic acid. (**D**) Citric acid. (**E**) Succinic acid. (**F**) Oxalic acid. (**G**) Acetic acid.

**Figure 6 foods-12-02715-f006:**
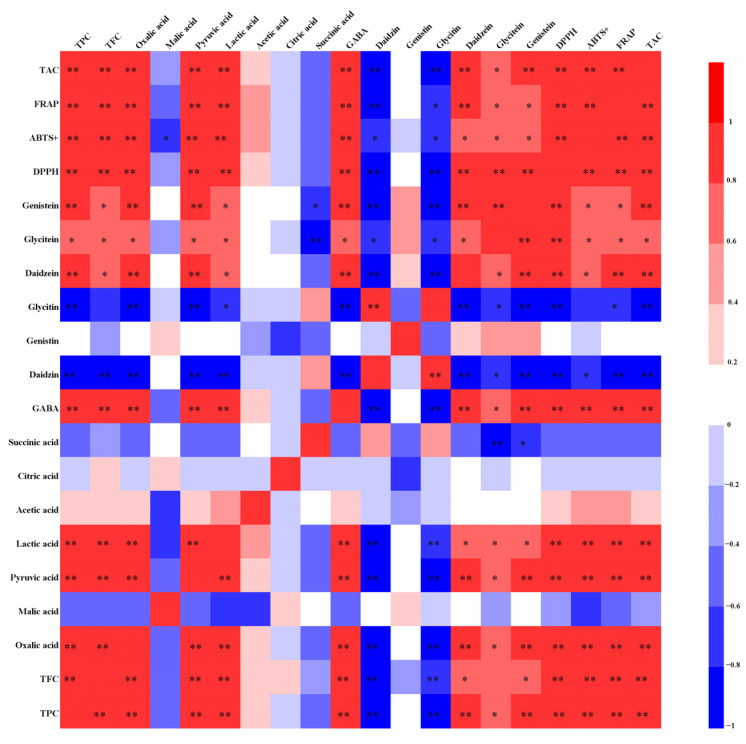
The correlation analysis between various active substances and antioxidant capacity. *, *p* < 0.05; **, *p* < 0.01.

**Table 1 foods-12-02715-t001:** TPC, TFC, and antioxidant activity for different fermented soymilks.

Group	TPC (mg/g)	TFC (mg/g)	DPPH Radical-Scavenging Rate/%	ABTS Radical-Scavenging Rate/%	Iron Ion Chelating Ability/%	Total Antioxidant Properties (mg/g)
PP-FSM	2.59 ± 0.028 ^a^	2.41 ± 0.28 ^a^	57.16 ± 0.33 ^b^	45.58 ± 0.72 ^a^	40.89 ± 0.78 ^a^	257.93 ± 1.25 ^b^
P-FSM	2.60 ± 0.057 ^a^	2.10 ± 0.034 ^b^	57.63 ± 0.31 ^a^	42.27 ± 0.32 ^b^	38.82 ± 0.28 ^b^	263.4 ± 1.67 ^a^
FSM	2.04 ± 0.023 ^b^	1.86 ± 0.074 ^c^	52.67 ± 0.13 ^c^	34.83 ± 0.31 ^c^	37.43 ± 0.59 ^c^	215.17 ± 0.98 ^c^

^a–c^ Means within three samples with different superscripts are significantly different (*p* < 0.05).

## Data Availability

The data presented in this study are available on request from the corresponding author.

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
