# Peer review of "Effects of Polygonatum sibiricum on Physicochemical Properties, Biological Compounds, and Functionality of Fermented Soymilk"

_foods, 2023, doi:10.3390/foods12142715_

Round 1

Reviewer 1 Report

This manuscript discussed the potential of producing soy-based milk fermented products enriched with P.sibiricum. Several aspect on its functional properties were highlighted in this study. However, there are few suggestion to improve the manuscript.

Overall: What are the targeted end product by author? Will it be yogurt-like or cultured drink-like product? Also, why author used P.sibiricum polysaccharides? Suggest author to highlight this in abstract / introduction.

Introduction:

-          Line 33 – error for references

Material & method:

-          Line 85: why cellulose & papain were added? Any target compound? Can this be considered as crude extract? Suggest author to explain a bit on this.

-          Line 105 – 106: at what concentration P.sibiricum were added? Please include

-          Line 108 – 109: sentence not completed

-          Line 144 -152: include details of HPLC condition, the name of column, flowrate, detector, etc.

Result & discussion: well explained and discussed by the author.

-          Line 268 -269 :  how can you confirm your statement on inhibition of E.coli & c. perfingen as no result were shown in this manuscript

Suggest for professional english proficiency proof read especially on abstract & introduction section.

Author Response

Dear reviewer :

Thank you very much for your highly insightful and thoughtful advice. We deeply appreciate the time and effort you’ve spent in reviewing our manuscript. Your instructive comments are really thoughtful and helpful. We convinced that your constructive comments have made a great improvement on our manuscript. We had revised and rechecked all problems you had mentioned. Based on your comments, our description of the revision and the revised original text are shown in the attachment.

Reviewer 2 Report

The manuscript “Effects of Polygonatum sibiricum on physicochemical, biological compounds and functionality of fermented soymilk” by Liut et al. investigated the use of the medicinal plant in fermented soymilk.

Overall, this paper shows interesting data that fits into the topic of the journal, However, I have a few comments that the authors should give consideration.

The statistical analysis: The authors used a One-way ANOVA to compare the 3 groups (i.e Figures 1 & 2) over time, the authors needed to use TWO-ways ANOVA, to compare the fermentation (group) effect as well as the time effect.

The viability of Lactobacilli in the fermented product is not well described in the methodology section. The ratio lactobacilli over total bacteria in each product over time need to be included. This will show the proportion of lactobacilli load in the product during the fermentation and storage and how P. sibiricum is affecting this ratio.
The correlation analysis is not very well explained; please add a description (and statistical method) in the methodology section.

Abbreviations in the abstract need to be explained.

Reference 1 in the introduction sections needed to be verified

Author Response

Dear reviewer :

Thank you very much for your highly insightful and thoughtful advice. We deeply appreciate the time and effort you’ve spent in reviewing our manuscript. Your instructive comments are really thoughtful and helpful. We convinced that your constructive comments have made a great improvement on our manuscript. Based on your comments, our description of the revision and the revised original text are shown in the attachment.
